# Basic biology education in high school and acceptance of genetically modified food in Japan

**Akihiro Mine**[1][*], **Sawako Okamoto**[1], **Tomoya Myojin**[1][*], **Miki Hamada**[2], **Tomoaki Imamura**[1]

1 Department of Public Health, Health Management and Policy, Nara Medical University, Kashihara, Nara, Japan, 2 Advanced Service Development Group, Innovation Service Creation Division, Mitsubishi Research Institute, Inc., Tokyo, Japan

☯ These authors contributed equally to this work.
* akihiromn777@gmail.com (AM); t-myojin@naramed-u.ac.jp (TM)

**Data Availability Statement:** All data used in this study are within the paper and its Supporting information files.

## Abstract

While many types of genetically modified (GM) food have become more available worldwide, the acceptance of GM food continues to be low. To increase this acceptance, various educational interventions have been conducted; however, conflicts remain about the safety and acceptability of GM food among laypeople, experts, and policymakers in several countries. Thus, this study aimed to clarify whether basic biology education influences Japanese people's acceptance of GM food. We examined this idea by comparing individual experiences of high school biology education based on curriculum and proficiency level. We distributed online questionnaires to 1,594 people in Japan; 1,122 valid responses (70.4%) were obtained. Results showed that the acceptance rates of GM vegetables, fish, and meat were 33.6%, 29.0%, and 29.1%, respectively. Comparing the biology knowledge test scores according to different high school biology education levels (i.e., non-learners, basic, and advanced levels) showed no significant differences between the three age groups (20s, 30s, and 40s), which corresponded to different curricula ($p = 0.90$); however, there were significant differences between the high school biology education levels ($p<0.01$). Using logistic regression analysis, we then examined the effect of high school biology education on acceptance of GM food. The results showed no significant differences between different high school biology education levels but significantly lower acceptance by the 30s and 40s groups compared with the 20s group ($p<0.05$). This suggests that basic biology education alone is not sufficient to change people's attitudes toward GM foods. These generational differences suggest factors other than high school biology curriculum affect different generations' acceptance of GM foods. Overall, high school biology education did not directly affect acceptance of GM foods, although those who received a higher level of education had an increased knowledge of GM foods.

**Funding:** This research was funded by a Health and Labor Sciences Research Grant (#H27-Food-General-004) in 2016. However, the funder had no role in study design, data collection and analysis, decision to publish, or preparation of the manuscript.

**Competing interests:** The authors have declared that no competing interests exist.

# Introduction

## Background and objective

Over the last few decades, the commercialization of genetically modified (GM) food has rapidly spread worldwide [1]. However, public acceptance of GM food has remained low [2–4], and conflict has arisen between experts and the public regarding the perception of GM food [5]. Particularly in European countries, such as France, Germany, and the United Kingdom, many people strongly refuse to consume GM food [2]. Even in the United States, which produces the most GM food worldwide, more than half of laypeople show low acceptance of it and refuse to consume it [4]. In Japan, GM foods, such as herbicide-tolerant soybeans and insect-resistant corn, appeared in 1996, and the labeling of such foods was mandated in 2001 [6]. In addition, before a new type of GM food is introduced on the market, it must undergo a food safety assessment from the Ministry of Health, Labour and Welfare [6]. However, despite these precautions, Japanese people seem to refuse GM food at similar levels to the aforementioned countries [3, 4, 7, 8].

Regarding the public acceptance of GM food, some studies have indicated that the public refused applications of new technology to foods because they did not have correct information [5]. Thus, various educational interventions aiming to provide laypeople with sufficient knowledge to select GM food have been examined. The results of these interventions are varied; some studies indicated that increasing knowledge affected consumers' acceptance of GM food [9–13], whereas others found that knowledge alone did not lead to an increase in laypeople's acceptance [14]. Thus, there is no consensus on the relationship between technical or other GM food-related knowledge and people's acceptance of GM food.

To understand GM food, it is important to possess basic biology knowledge, including knowledge of relevant technology and of how food is digested, which indicates that the genetic modification process is not harmful. In Japan, the high school enrollment rate is very high (98% in 2018) [15], and high school students can choose the biology course level they take. The contents of curricular subjects, including biology, are updated every 10 years; accordingly, the range and volume of educational content on GM technology and food in biology courses has increased with each update (S1 Table). Thus, the effects of different biology curricula and education levels on people's knowledge and acceptance of GM food can be compared. We hypothesized that as students progress through the basic educational levels and obtain more basic biology knowledge, their acceptance of GM food increases. Although education itself does not necessarily move people's intention in a specific direction, the gap between objective and subjective knowledge can create resistance toward GM food [16]; therefore, when people gain more objective knowledge through education, they may become more acceptant. Thus, this study aimed to clarify whether high school educational interventions influence laypeople's understanding of GM food and whether this understanding increases its acceptance in Japan.

The word "risk" is defined in this study as the possibility of adverse events to human health and the environment caused by the widespread use and consumption of GM foods. This study's novelty rests on its focus on the acceptance of GM foods in terms of compulsory or equivalent basic education, duration of education, and timing of education. To our knowledge, this is the first study to examine how basic biology education in high school affects acceptance of GM foods among Japanese individuals.

## Theoretical framework

To examine what affects the acceptance of GM food, we considered various factors, including basic biology education, sociodemographic characteristics, psychological factors, and

sociological factors, which seemed to be associated with acceptance based on previous studies [5, 17–24].

In addition to biology education, consumers' acceptance of GM food may be related to their perception of GM technology and food; this perception is affected by various psychological aspects [5, 18, 25]. Psychological or sociological factors that were determined to affect the acceptance of GM foods in previous studies need to be considered [5, 17]. Studies have indicated that laypeople assess food not only using knowledge but also through multidimensional factors ascribed to personal values [18]. This is particularly evident in the attitude model for GM food, which was developed by Christoph et al. [19], who combined sociodemographic characteristics and perception of risks with knowledge in their multivariable model. Other studies have suggested that the perceived risks and benefits of GM food could be the biggest impact factors for public attitude [20–22]. Furthermore, trusting information sources is critical as the public relies on information given by experts and the media when assessing the benefits and risks of GM food [23, 24]. Therefore, psychological factors, such as risk perception, benefit perception, and trust, were included in this study's analyses. Scholars in the sociology field have stressed that the consumption of food should be considered in an everyday context [5]. Thus, we took some sociological factors into consideration, such as cooking and shopping.

## Materials and methods

### Description of the study area

The total population of Japan was 125.5 million in 2021, which is the 11th highest in the world [26]. Moreover, Japan is experiencing declining birthrate along with population aging. Since 1997, the aged population (65 years old and over) have surpassed the youth population (0–14 years old) [26]. The average size of households decreased sharply in 1970, and in 2020, the average number of household members was 2.21 [26]. Japan is an island country situated on the Eurasian continent in the northern hemisphere with a total surface area of 377,974 square kilometers [26]. The archipelago of Japan has a temperate marine climate, which varies regionally based on seasonal winds and ocean currents [26]. The number of workers in the agricultural, forestry, and fishing industries in Japan has been decreasing annually; in 2020, only 2.13 million people (3.2% of the industrial worker population) worked in these industries [26]. Accompanying this, the food self-sufficiency ratio of Japan has been dropping and was 37% in fiscal year 2020 [26]. The self-sufficiency ratio of rice was 97%, whereas the ratio for wheat and beans were only 15% and 8%, respectively [26]. Therefore, Japan depends mostly on imports for the supply of those items, which are the primary imported GM foods in Japan. Finally, regarding Japan's educational system, its primary and secondary education is based on a 6-3-3 system [26]. There are nine years of compulsory schooling at the elementary and lower secondary school levels, and the rate of students pursue higher education is high. The high school enrollment rate was 98% in 2018 [15], and in 2021, 57.5% of upper secondary school and the upper division of secondary school graduates immediately entered an institution of higher learning, such as university or junior college [26].

### Data collection

The online questionnaire survey in this study was designed based on the results of research conducted over seven years [27–32]. Approximately 30 people participated in the pilot research for standardization in the first half of March 2016. These individuals were not involved in GM research and were not experts on GM. Then, the questionnaires were distributed and collected in Japan from March to April 2016 by Survey Research Center Co., Ltd., a private research company with a database of Japanese consumers who are interested in

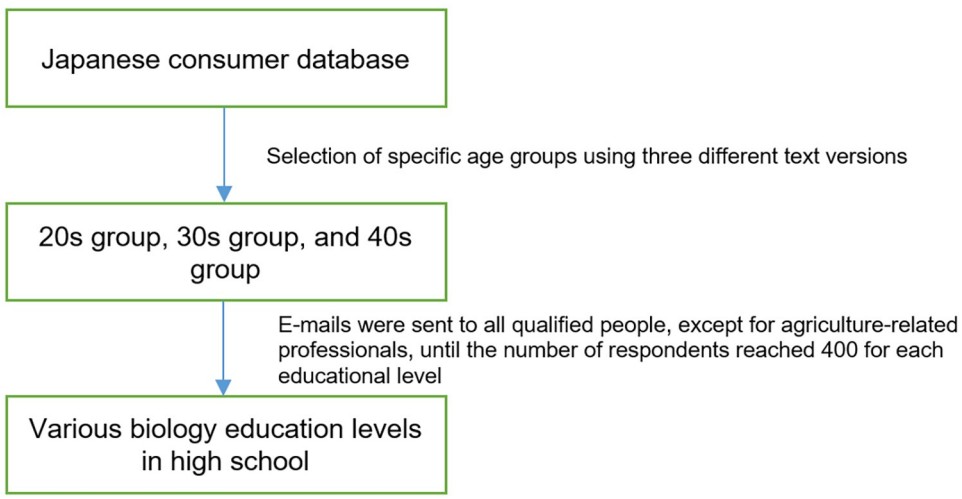

**Fig 1. Data collection procedure.**

participating in various surveys. We chose this company because it has a national reach and significant experience with online survey research. The company distributed the questionnaire to registrants and implemented a cutoff of 400 for enrollment in each of the three biology education level groups, the details of which are provided below (Fig 1). We have previously carried out a series of studies on the perception of GM food; in some of these studies, we determined a sample size of approximately 500 [27, 30]. In this study, the initial sample size was set at 500 for each education level, but the collection rate for the 20s group was poor. Seeking more responses by invitation e-mails would have resulted in a lower response rate and selection bias. Therefore, the target size was changed to 400. The total number of invitation e-mails was 1,594. After we excluded participants who did not respond completely and those who were in agriculture-related professions, such as farmers, fishermen, and researchers, 1,122 valid responses were obtained.

The survey items were as follows.

Firstly, to gather basic information on biology education, we collected data on the participants' ages to identify the biology education curriculum they studied, their biology education level in high school, and whether they had had an opportunity to study genetics, gene recombination, or biotechnology after graduating from high school. The biology curriculum for high school students differed depending on when the participant went to school; therefore, the first year and middle year of each generation were selected: a 20s group, comprising those aged 20 or 21 and 25 years and who used the textbook version that was in use from 2003 to 2012; a 30s group comprising those aged 30 and 35 years and who used a textbook version in use from 1994 to 2003; and a 40s group comprising those aged 40 and 45 years and who used a textbook version in use from 1982 to 1994. We only collected data from those with the lowest and median ages within the 10-year groups rather than from people of all ages from the 10-year range to reduce deviations related to response rates for various specific ages. Furthermore, the participants were divided into three groups according to their high school biology education level: the non-learner group (those who answered "non-scientific course" or "scientific course but didn't study biology"), basic-level group (those who answered "studied biology level 1"), and advanced-level group (those who answered "studied biology level 2" or "studied both biology levels 1 and 2"). We also analyzed the content of the textbooks used for each of the three

generations in the study. Although teachers in Japan are able to choose their textbooks, all text-books must follow a standardized government curriculum. We chose to use textbooks pub-lished by Tokyo Shoseki Co., Ltd. that followed this curriculum (S1 Table).

Secondly, to examine participants' current knowledge of GM technology, participants took a test consisting of nine items: six items about basic knowledge related to DNA and three items about digestion (S2 Table).

Thirdly, regarding sociodemographic characteristics, we collected information on sex, household income (divided into 11 groups by ¥1 million [US$1 = ¥137.6 as of November 13, 2022]), age, final education level, residence, and whether they have children living with them and the age groups of their children: under 6 years, from 6 to 11 years, or from 12 to 19 years. When respondents had more than one child, we counted their youngest child.

Fourthly, as practices of everyday life, we asked them whether they shopped for their daily food and cooked for themselves.

Fifthly, as psychological items, we asked participants about their trust in information sources and their perception of risks and benefits of GM food. Regarding the former, partici-pants were asked how much they trusted food-related information from the following sources: television news, infotainment television programs, web news, social networking services, pub-lic institutions, experts, producer groups, consumer groups, food companies, newspapers, magazines, books, and families or acquaintances. Regarding the latter, participants were asked about their degree of agreement with the following: "My nutritional condition is improved by enriched GM food"; "I can buy more affordable food thanks to GM technology"; "I can eat enough food thanks to the supply of GM food"; "I can eat foods that are pesticides-free thanks to GM technology"; "GM technology reduces negative effects on the environment around me"; "I can have a secure food supply thanks to GM technology"; "I feel GM food is unnatu-ral"; "I doubt the safety of GM food because it is new"; "GM food has uncertain risks to me"; "I feel that the government's safety inspection of GM food is insufficient"; and "I think GM food will cause health risks." Participants responded on a four-point Likert scale; 1 indicated "don't trust" or "don't agree," and 4 indicated "trust" or "agree," depending on the question type.

Lastly, regarding acceptance of GM food, we asked participants about their intention of eat-ing or resistance to eating GM foods, including GM vegetables, fish, and meat. Participants rated their intention to eat and resistance on a four-point Likert-type scale; 1 indicated "eat" or "no resistance," and 4 indicated "don't eat" or "have resistance," depending on the question type.

The questionnaire on the acceptance of GM food, trust in information sources, and percep-tion of risks and benefits was designed with four responses to avoid interim answers; these were converted into two values, positive and negative, in the statistical analysis. For trust in information sources, perception of risks, and perception of benefits, responses of 1 and 2 were converted to 0 ("don't trust," "don't agree," "don't eat," or "no resistance"), and responses of 3 and 4 were converted to 1 ("trust," "agree," "eat," or "have resistance").

## Data analysis

First, to clarify the differences in current knowledge based on education, we compared respon-dents' biology knowledge test scores according to their high school biology education level by performing a one-way analysis of variance (ANOVA). ANOVA is used for cases of a quantita-tive outcome with a categorical explanatory variable [33]. Second, binominal logistic analysis was conducted to assess the impact of the various factors, such as demographics, basic biology education, trust in information sources, perception of risks, perception of benefits, and prac-tices of everyday life, on acceptance of GM food. Logistic analysis was used because it predicts

the odds of an event outcome from a set of predictors and is suitable for dichotomous dependent variables [34]. Before the binominal logistic analysis, we conducted a principal component analysis (PCA). We chose the items with the highest principal component scores as the independent factors to substitute for psychological factors in the binominal logistic analysis, as psychological factors might have multicollinearity in logistic analysis. Additionally, to ensure the robustness of our analyses, another analysis without risk and benefit perception was conducted, as risk and benefit perceptions have a strong impact on acceptance [20–22]. All statistical analyses were conducted using IBM SPSS version 21.

This study was approved by the Ethics Committee of Nara Medical University, with the authorization code 655. Participants provided written informed consent before they answered the questionnaire. They were informed that their responses would be anonymous and only used for research purposes, that they could withdraw consent at any time during the questionnaire, and that after submitting their responses, they could not withdraw them because they would be anonymized. If they agreed with these instructions, they clicked a button to proceed to the questionnaire.

## Results

### Sociodemographic characteristics

The questionnaire was distributed online to 1,594 people, and 1,122 valid responses (70.4%) were obtained (Table 1). We excluded participants with incomplete or missing responses and

**Table 1. Participants' sociodemographic characteristics (n = 1,122).**

| Variable | | | n | % | Overall Japanese population |
|---|---|---|---|---|---|
| **Sex** | Male | | 526 | 46.9 | 50.7[a] |
| | Female | | 596 | 53.1 | 49.3[a] |
| **Age** | 20s group | 20 or 21 | 183 | 16.3 | |
| | | 25 | 171 | 15.2 | |
| | 30s group | 30 | 181 | 16.1 | |
| | | 35 | 187 | 16.7 | |
| | 40s group | 40 | 194 | 17.3 | |
| | | 45 | 206 | 18.4 | |
| **Income** | Median household income | | ¥500–600 million | | ¥427 million[b] |
| **Education** | High school degree | | 170 | 15.2 | |
| | Junior college or technical college diploma | | 228 | 20.3 | |
| | University degree, humanities | | 411 | 36.6 | |
| | University degree, science | | 221 | 19.7 | |
| | Graduate degree, humanities | | 33 | 2.9 | |
| | Graduate degree, science | | 59 | 5.3 | |
| **Children** | No children in the household | | 775 | 69.1 | |
| | Under 6 years old | | 202 | 18.0 | 4.8[c] |
| | 6 to 11 years old | | 85 | 7.6 | 21.0[c] |
| | 12 to 19 years old | | 60 | 5.3 | 14.0[c] |
| **Residence** | Rural | | 454 | 40.5 | 48.0[c] |
| | Urban | | 668 | 59.5 | 52.0[c] |

[a]2015 Population Census, only people aged 20 to 49 were extracted

[b]2016 Comprehensive Survey of Living Conditions.

[c]2015 Population Census, all households.

those who were in agriculture-related professions, such as farmers, fishermen, and researchers. The data collection is shown procedure in Fig 1. The ratio of male to female participants was larger than the ratio of male to female individuals in Japan in the relevant age group, based on the 2015 Population Census [35]. The median household income was ¥5–6 million, which was more than the ¥4.27 million average income in the 2016 Comprehensive Survey of Living Conditions [36]. The percentage of those who had children in this study was larger than the data indicated for the Japanese population in the 2015 Population Census [35].

## PCA of psychological factors

PCA was conducted for questions about trust in information sources and for the perception of risks and benefits. For the PCA results, we identified four principal components: "trust in public information," "trust in unofficial information," "perception of benefits," and "perception of risks." The Cronbach's α for these four components were 0.91, 0.84, 0.94, and 0.94, respectively (Table 2). The sample size was suitable because the value for the Bartlett test of sphericity was less than 0.01 and that for the Kaiser–Meyer–Olkin measure of sampling adequacy was 0.92. We extracted the items that had the largest main component loading from each of the four principal components and used them in the binominal logistic analysis.

## Differences in textbook content according to high school biology curriculum and level

S1 Table shows that the advanced-level groups in all three age groups were taught basic information regarding genetics and DNA; however, the 40s age group did not learn about the mechanisms of DNA replication or gene recombination in high school, in contrast to the 20s and 30s groups.

## Differences in genetic modification and digestion knowledge

The biology knowledge test results were analyzed using ANOVA. There were no statistically significant differences between the biology knowledge scores of the three age groups ($F = 0.10$, $p = 0.90$); however, there were significant differences among the levels of high school biology education ($F = 34.12$, $p<0.01$; Table 3). The results of Tukey's tests showed that those with a higher level of biology education tended to obtain higher scores on the biology knowledge test (Table 4). The distribution of biology test scores by biology education level are shown in S1 Fig.

## Acceptance of GM food based on various factors

The acceptance rate of GM food according to various factors is shown in Table 5. Respondents' overall acceptance rates of GM vegetables, fish, and meat were 33.6%, 29.0%, and 29.1%, respectively. The resistance rate was 55.7% Acceptance rates of GM vegetables were greater than for GM fish and meat, and those of GM fish and meat were similar for all groups.

**Differences in GM food acceptance between different age groups and biology education levels.** The logistic regression analysis results regarding GM food acceptance conducted to examine the effect of high school biology education are shown in Table 6. Regarding validity, according to the Nagelkerke $R^2$ value, the model explained 26%–29% of variance, and predictions were accurate in 69%–75% of cases. Using the 20s group as a reference, we found that the various acceptance barometers in the 30s and 40s groups were significantly low. The 30s group had a significantly low intention to eat GM food ($p<0.05$), and the 40s group had a significantly low intention to eat GM food and high resistance ($p<0.05$). However, there was no

**Table 2. Promax rotated component loadings for trust in information sources and perception of risks and benefits of GM food.**

| | Trust in public information α = 0.907 | Perception of benefits α = 0.939 | Perception of risks α = 0.936 | Trust in unofficial information α = 0.843 |
|---|---|---|---|---|
| Trust in information about food from public institutions. | **0.98** | 0.00 | 0.00 | -0.27 |
| Trust in information about food from producers' groups. | **0.94** | -0.04 | 0.02 | -0.15 |
| Trust in information about food from consumers' groups. | **0.88** | -0.05 | 0.10 | -0.06 |
| Trust in information about food from experts. | **0.83** | 0.03 | 0.00 | -0.06 |
| Trust in information about food from newspapers. | **0.68** | 0.02 | -0.02 | 0.18 |
| Trust in information about food from television news. | **0.65** | 0.03 | -0.05 | 0.23 |
| Trust in information about food from food companies. | **0.63** | 0.01 | -0.10 | 0.16 |
| I can eat enough food thanks to the supply of GM food. | 0.00 | **0.92** | -0.01 | -0.01 |
| I can eat foods that are pesticide-free thanks to GM technology. | 0.01 | **0.90** | 0.01 | -0.02 |
| I can have a secured food supply thanks to GM technology. | 0.02 | **0.90** | 0.02 | -0.06 |
| My nutritional condition is improved by enriched GM food. | -0.03 | **0.85** | -0.06 | 0.05 |
| GM technology reduces negative effects on the environment around me. | -0.05 | **0.85** | 0.02 | 0.04 |
| I can buy more affordable food thanks to GM technology. | 0.03 | **0.84** | 0.04 | -0.02 |
| GM food has uncertain risks to me. | 0.05 | 0.03 | **0.91** | -0.04 |
| I doubt the safety of GM food because it is new. | 0.06 | 0.02 | **0.91** | -0.06 |
| I feel that the government's safety inspection of GM food is insufficient. | -0.08 | 0.04 | **0.90** | 0.04 |
| I think GM food will cause health risks. | -0.08 | -0.06 | **0.88** | 0.09 |
| I feel GM food is unnatural. | 0.04 | -0.01 | **0.84** | 0.04 |
| Trust in information about food from food social networking services. | -0.28 | 0.02 | -0.03 | **0.93** |
| Trust in information about food from food magazines. | 0.15 | -0.02 | 0.03 | **0.74** |
| Trust in information about food from food families or acquaintances. | 0.04 | -0.05 | 0.11 | **0.62** |
| Trust in information about food from food books. | 0.31 | -0.04 | 0.07 | **0.56** |
| Trust in information about food from food infotainment television programs. | 0.40 | 0.03 | -0.08 | **0.47** |
| Trust in information about food from web news. | 0.38 | 0.05 | -0.02 | **0.46** |

GM: genetically modified.

significant difference in intention to eat GM food and in resistance to GM food among high school biology education levels.

**Effect of factors other than high school biology education on increasing acceptance of GM food.** The results of the logistic regression analysis regarding GM food acceptance, conducted to examine the effect of factors other than basic biology education, are shown in Table 7. Regarding sociodemographic characteristics, factors that affected GM food acceptance were sex and income. Female participants showed significantly low acceptance of GM food

**Table 3. Level of high school biology education and biology knowledge test score.**

| Biology education level in high school | n | Mean score (95% CI) |
|---|---|---|
| Non-learner group | 396 | 5.69 (5.47 to 5.91) |
| Basic-level group | 381 | 6.50 (6.30 to 6.69) |
| Advanced-level group | 345 | 6.90 (6.70 to 7.09) |

High school biology education levels were divided into three groups: the non-learner group (those who answered "non-scientific course" or "scientific course but didn't study biology"), basic-level group (those who answered "studied biology level 1"), and advanced-level group (those who answered "studied biology level 2" or "studied both biology levels 1 and 2").

($p<0.01$), as did those with higher household income ($p<0.01$). Acceptance according to educational background did not demonstrate any significant differences. Regarding psychological factors, those who trusted unofficial information showed higher acceptance of GM food and GM meat ($p<0.01$). Perception of benefits and risks significantly affected all outcomes ($p<0.01$). As for everyday routine, those who did their own shopping showed a lower acceptance of GM meat ($p<0.05$).

## Additional analysis of the effect of factors on increasing acceptance of GM food

Finally, the results of additional analysis excluding risk and benefit perceptions are shown in Table 8. Regarding validity, according to the Nagelkerke $R^2$ value, the model explained 10%–14% of the variance, and predictions were accurate in 65%–72% of cases. A trend similar to that in the main analysis was found regarding acceptance differences according to age groups and education level in this additional analysis. Compared with the 20s group, the 30s and 40s groups had significantly lower intentions to eat all GM foods ($p<0.05$). Different high school biology education levels showed no significant difference in intention or resistance to eat GM foods. Regarding other factors except basic biology education, the sociodemographic factors of female sex and higher household income showed significantly lower acceptance of GM food ($p<0.01$). Those who trusted public information showed higher acceptance of GM vegetables ($p<0.01$), and those who trusted unofficial information showed higher acceptance of GM fish and meat ($p<0.01$). In contrast, those who went shopping by themselves every day showed significantly high resistance to all GM foods. This trend was not observed in the first analysis that included risk and benefit perceptions.

**Table 4. Tukey's test results for level of high school biology education and knowledge test score.**

| (A) Biology education level | (B) Biology education level | Mean difference between knowledge test scores (95% CI) (A)-(B) | p value |
|---|---|---|---|
| Advanced-level group | Basic-level group | 0.40 (0.04 to 0.75) | 0.23 |
| | Non-learner group | 1.21 (0.85 to 1.56) | <0.01 |
| Basic-level group | Non-learner group | 0.81 (0.46 to 1.15) | <0.01 |

High school biology education levels were divided into three groups: the non-learner group (those who answered "non-scientific course" or "scientific course but didn't study biology"), basic-level group (those who answered "studied biology level 1"), and advanced-level group (those who answered "studied biology level 2" or "studied both biology levels 1 and 2").

**Table 5. Acceptance rates of GM food according to biology education, sociodemographic characteristics, trust in institutions, perception of benefits and risks, and practices of everyday life.**

| Intention to eat or resistance toward | GM vegetable | GM fish | GM meat | Resistance |
|---|---|---|---|---|
| | rate (%) | | | |
| **All respondents** | 33.6 | 29.0 | 29.1 | 55.7 |
| **Basic biology education** | | | | |
| Age (educational curriculum) | | | | |
| 20s group | 43.2 | 39.0 | 40.4 | 44.9 |
| 30s group | 30.4 | 25.5 | 25.0 | 54.3 |
| 40s group | 28.0 | 23.3 | 23.0 | 66.5 |
| Biological education level in high school | | | | |
| Non-learner group | 36.9 | 32.1 | 32.6 | 52.0 |
| Basic-level group | 31.5 | 26.8 | 26.5 | 58.3 |
| Advanced-level group | 32.2 | 27.8 | 28.1 | 57.1 |
| Opportunity to study after high school | | | | |
| No | 32.4 | 28.1 | 28.0 | 56.4 |
| Yes | 38.2 | 32.2 | 33.5 | 53.2 |
| **Other factors** | | | | |
| Sex | | | | |
| Male | 40.1 | 35.9 | 36.1 | 47.5 |
| Female | 27.9 | 22.8 | 23.0 | 62.9 |
| Household income (by ¥1 million) | | | | |
| under ¥1 million | 48.2 | 44.6 | 45.8 | 42.2 |
| ¥1 million–under 2 million | 44.4 | 40.7 | 37.0 | 29.6 |
| ¥2 million–under 3 million | 37.8 | 29.7 | 31.1 | 50.0 |
| ¥3 million–under 4 million | 39.6 | 35.8 | 36.6 | 47.0 |
| ¥4 million–under 5 million | 34.6 | 29.0 | 30.2 | 54.9 |
| ¥5 million–under 6 million | 31.7 | 26.6 | 25.9 | 56.1 |
| ¥6 million–under 7 million | 33.6 | 30.7 | 29.9 | 62.0 |
| ¥7 million–under 8 million | 33.0 | 27.3 | 28.4 | 59.1 |
| ¥8 million–under 9 million | 25.6 | 20.7 | 20.7 | 62.2 |
| ¥9 million–under 10 million | 24.2 | 21.0 | 19.4 | 62.9 |
| ¥10 million or more | 24.6 | 20.1 | 20.1 | 65.7 |
| Final education level | | | | |
| High school degree | 42.4 | 35.9 | 36.5 | 48.8 |
| Junior college or technical college diploma | 29.4 | 24.6 | 24.6 | 56.6 |
| University degree, humanities | 29.4 | 25.8 | 26.8 | 61.8 |
| University degree, science | 36.2 | 33.0 | 32.1 | 51.6 |
| Graduate degree, humanities | 30.3 | 18.2 | 18.2 | 54.5 |
| Graduate degree, science | 45.8 | 39.0 | 37.3 | 45.8 |
| Residential area | | | | |
| Rural | 32.8 | 27.5 | 27.8 | 55.9 |
| Urban | 34.1 | 29.9 | 30.1 | 55.5 |
| Children living in household | | | | |
| No children | 36.5 | 31.5 | 31.5 | 53.0 |
| Children under 6 years old | 27.2 | 24.3 | 25.2 | 56.4 |
| Children 6–11 years old | 23.5 | 20.0 | 21.2 | 74.1 |
| Children 12–19 years old | 31.7 | 25.0 | 23.3 | 61.7 |
| Trust in public information | | | | |

*(Continued)*

**Table 5.** (Continued)

| Intention to eat or resistance toward | GM vegetable | GM fish | GM meat | Resistance |
|---|---|---|---|---|
| | rate (%) | | | |
| Don't trust | 27.1 | 26.8 | 25.7 | 59.9 |
| Trust | 35.8 | 29.7 | 30.3 | 54.3 |
| Trust in unofficial information | | | | |
| Don't trust | 32.1 | 25.7 | 25.7 | 56.9 |
| Trust | 37.6 | 37.6 | 38.2 | 52.6 |
| Perception of benefits | | | | |
| Don't agree | 15.0 | 13.1 | 13.5 | 74.3 |
| Agree | 48.8 | 42.0 | 42.0 | 40.5 |
| Perception of risks | | | | |
| Don't agree | 37.4 | 35.1 | 36.0 | 45.6 |
| Agree | 31.9 | 26.3 | 26.2 | 60.1 |
| Shopping in everyday life | | | | |
| Yes | 44.8 | 42.1 | 42.1 | 42.8 |
| No | 31.9 | 27.0 | 27.2 | 57.6 |
| Cooking in everyday life | | | | |
| By others | 35.8 | 30.2 | 30.0 | 53.1 |
| For themselves | 31.8 | 27.9 | 28.4 | 57.8 |

GM: genetically modified.

## Discussion

The purpose of this study was to investigate whether basic biology education in Japanese high schools influences the acceptance of GM foods. The results show that those who had received higher levels of biology education in high school had more knowledge of GM technology; however, a higher level of knowledge did not create a stronger intention to eat GM foods (i.e., acceptance). Previous studies examining the impact of education on acceptance of GM foods has reported various results. Some found that gaining knowledge through education increases

**Table 6. Odds ratios of logistic regression analysis explaining acceptance of GM food according to high school biology education.**

| Intention to eat or resistance toward | GM vegetable | GM fish | GM meat | Resistance |
|---|---|---|---|---|
| | Odds ratio (95% CI) | | | |
| Age (educational curriculum) | | | | |
| 20s group | (ref) | (ref) | (ref) | (ref) |
| 30s group | 0.64 (0.45 to 0.92)* | 0.56 (0.38 to 0.82)** | 0.50 (0.34 to 0.73)** | 1.37 (0.97 to 1.95) |
| 40s group | 0.67 (0.46 to 0.98)* | 0.60 (0.40 to 0.89)* | 0.54 (0.36 to 0.80)** | 2.00 (1.38 to 2.89)** |
| Biological education level in high school | | | | |
| Non-learner group | (ref) | (ref) | (ref) | (ref) |
| Basic-level group | 0.92 (0.65 to 1.30) | 0.92 (0.65 to 1.32) | 0.86 (0.60 to 1.23) | 1.07 (0.77 to 1.50) |
| Advanced-level group | 0.88 (0.61 to 1.26) | 0.90 (0.62 to 1.31) | 0.90 (0.62 to 1.30) | 1.04 (0.73 to 1.47) |
| Opportunity to study after high school | | | | |
| No | (ref) | (ref) | (ref) | (ref) |
| Yes | 1.16 (0.79 to 1.70) | 1.00 (0.67 to 1.49) | 1.12 (0.75 to 1.67) | 1.05 (0.72 to 1.54) |

** and * report statistical significance at the 0.01 and 0.05 levels, respectively. (ref) means reference group for statistical comparison. GM: genetically modified.

**Table 7. Odds ratios of logistic regression analysis explaining acceptance of GM food according to sociodemographic characteristics, trust in institutions, perception of benefits and risks, and practices of everyday life.**

| Intention to eat or resistance toward | GM vegetable | GM fish | GM meat | Resistance |
|---|---|---|---|---|
| | Odds ratio (95% CI) | | | |
| Sex | | | | |
| Male | (ref) | (ref) | (ref) | (ref) |
| Female | 0.61 (0.45 to 0.84)** | 0.54 (0.39 to 0.74)** | 0.51 (0.37 to 0.71)** | 1.96 (1.45 to 2.64)** |
| Household income (by ¥1 million) | 0.93 (0.88 to 0.98)** | 0.93 (0.88 to 0.98)** | 0.92 (0.87 to 0.97)** | 1.08 (1.03 to 1.14)** |
| Final education level | | | | |
| High school degree | (ref) | (ref) | (ref) | (ref) |
| Junior college or technical college diploma | 0.59 (0.37 to 0.94)* | 0.61 (0.38 to 0.98)* | 0.59 (0.37 to 0.97)* | 1.27 (0.81 to 1.99) |
| University degree, humanities | 0.61 (0.40 to 0.95)* | 0.64 (0.41 to 1.00) | 0.65 (0.42 to 1.02) | 1.68 (1.10 to 2.56)* |
| University degree, science | 0.59 (0.35 to 0.98)* | 0.70 (0.42 to 1.18) | 0.61 (0.36 to 1.02) | 1.41 (0.86 to 2.33) |
| Graduate degree, humanities | 0.51 (0.19 to 1.36) | 0.40 (0.14 to 1.15) | 0.40 (0.14 to 1.13) | 1.06 (0.43 to 2.59) |
| Graduate degree, science | 1.01 (0.48 to 2.12) | 0.97 (0.47 to 2.02) | 0.82 (0.39 to 1.73) | 1.00 (0.48 to 2.07) |
| Residential area | | | | |
| Rural | (ref) | (ref) | (ref) | (ref) |
| Urban | 1.05 (0.78 to 1.41) | 1.14 (0.84 to 1.54) | 1.14 (0.84 to 1.55) | 0.95 (0.72 to 1.26) |
| Children living in household | | | | |
| No children | (ref) | (ref) | (ref) | (ref) |
| Children under 6 years old | 0.78 (0.53 to 1.17) | 0.88 (0.58 to 1.32) | 0.97 (0.65 to 1.47) | 0.96 (0.66 to 1.38) |
| Children 6–11 years old | 0.68 (0.37 to 1.27) | 0.79 (0.42 to 1.50) | 0.91 (0.48 to 1.72) | 1.6 (0.89 to 2.89) |
| Children 12–19 years old | 0.77 (0.39 to 1.49) | 0.82 (0.41 to 1.62) | 0.75 (0.37 to 1.50) | 1.16 (0.62 to 2.17) |
| Trust in public information | | | | |
| Don't trust | (ref) | (ref) | (ref) | (ref) |
| Trust | 1.11 (0.92 to 1.33) | 0.96 (0.68 to 1.37) | 1.08 (0.76 to 1.53) | 0.76 (0.55 to 1.05) |
| Trust in unofficial information | | | | |
| Don't trust | (ref) | (ref) | (ref) | (ref) |
| Trust | 0.98 (0.80 to 1.20) | 1.68 (1.22 to 2.31)** | 1.72 (1.25 to 2.36)** | 1.04 (0.77 to 1.41) |
| Perception of benefits | | | | |
| Don't agree | (ref) | (ref) | (ref) | (ref) |
| Agree | 3.86 (3.05 to 4.89)** | 6.27 (4.39 to 8.96)** | 6.16 (4.31 to 8.81)** | 0.17 (0.12 to 0.23)** |
| Perception of risks | | | | |
| Don't agree | (ref) | (ref) | (ref) | (ref) |
| Agree | 0.45 (0.36 to 0.56)** | 0.38 (0.27 to 0.54)** | 0.36 (0.35 to 0.54)** | 3.49 (2.49 to 4.88)** |
| Shopping in everyday life | | | | |
| No | (ref) | (ref) | (ref) | (ref) |
| Yes | 0.86 (0.56 to 1.34) | 0.65 (0.42 to 1.01) | 0.64 (0.41 to 1.00)* | 1.28 (0.84 to 1.96) |
| Cooking in everyday life | | | | |
| By others | (ref) | (ref) | (ref) | (ref) |
| By themselves | 0.97 (0.69 to 1.37) | 1.33 (0.93 to 1.90) | 1.40 (0.98 to 2.00) | 0.91 (0.66 to 1.26) |

** and * indicate statistical significance at the 0.01 and 0.05 levels, respectively. (ref) means reference group for statistical comparison. GM: genetically modified.

acceptance of GM foods and technologies [9–13, 37, 38], whereas others determined that increasing knowledge through education does not affect acceptance of GM foods and technologies [14]. Few other studies reported that increasing knowledge through education strengthens preexisting attitudes toward GM foods rather than changing them [19, 20, 39]. Thus, the results of our study indicate that high school biology education neither negatively nor positively affects acceptance of GM foods.

**Table 8. Odds ratios of logistic regression analysis explaining acceptance of GM food according to basic biology education, sociodemographic characteristics, trust in institutions, and practices of everyday life.**

| Intention to eat or resistance toward | GM vegetable | GM fish | GM meat | Resistance |
|---|---|---|---|---|
| | Odds ratio (95% CI) | | | |
| **Basic biology education** | | | | |
| Age (educational curriculum) | | | | |
| 20s group | (ref) | (ref) | (ref) | (ref) |
| 30s group | 0.67 (0.48 to 0.94)* | 0.58 (0.41 to 0.83)** | 0.52 (0.36 to 0.74)** | 1.39 (1.01 to 1.93)* |
| 40s group | 0.60 (0.43 to 0.86)** | 0.55 (0.38 to 0.79)** | 0.50 (0.35 to 0.72)** | 2.17 (1.55 to 3.05)** |
| Biology education level in high school | | | | |
| Non-learner group | (ref) | (ref) | (ref) | (ref) |
| Basic-level group | 0.89 (0.65 to 1.23) | 0.89 (0.64 to 1.25) | 0.84 (0.60 to 1.17) | 1.11 (0.82 to 1.52) |
| Advanced-level group | 0.92 (0.66 to 1.29) | 0.94 (0.66 to 1.33) | 0.93 (0.65 to 1.32) | 1.04 (0.75 to 1.43) |
| Opportunity to study after high school | | | | |
| No | (ref) | (ref) | (ref) | (ref) |
| Yes | 1.19 (0.83 to 1.70) | 1.05 (0.72 to 1.53) | 1.17 (0.80 to 1.70) | 1.01 (0.71 to 1.43) |
| **Factors except basic biology education** | | | | |
| Sex | | | | |
| Male | (ref) | (ref) | (ref) | (ref) |
| Female | 0.55 (0.41 to 0.74)** | 0.5 (0.37 to 0.68)** | 0.48 (0.35 to 0.65)** | 2.07 (1.57 to 2.74)** |
| Household income (by ¥1 million) | 0.92 (0.87 to 0.96)** | 0.91 (0.87 to 0.96)** | 0.91 (0.86 to 0.96)** | 1.10 (1.05 to 1.15)** |
| Final education level | | | | |
| High school degree | (ref) | (ref) | (ref) | (ref) |
| Junior college or technical college diploma | 0.59 (0.38 to 0.92)* | 0.61 (0.39 to 0.96)* | 0.60 (0.38 to 0.95)* | 1.25 (0.82 to 1.91) |
| University degree, humanities | 0.52 (0.35 to 0.78)** | 0.58 (0.38 to 0.89)* | 0.59 (0.39 to 0.90)* | 1.80 (1.22 to 2.67)** |
| University degree, science | 0.58 (0.36 to 0.93)* | 0.7 (0.43 to 1.14) | 0.61 (0.37 to 1.00)* | 1.41 (0.89 to 2.24) |
| Graduate degree, humanities | 0.58 (0.25 to 1.34) | 0.42 (0.16 to 1.13) | 0.42 (0.16 to 1.12) | 1.20 (0.54 to 2.66) |
| Graduate degree, science | 1.03 (0.53 to 1.97) | 1.1 (0.56 to 2.17) | 0.96 (0.49 to 1.91) | 0.92 (0.48 to 1.78) |
| Residential area | | | | |
| Rural | (ref) | (ref) | (ref) | (ref) |
| Urban | 1.15 (0.88 to 1.51) | 1.19 (0.90 to 1.59) | 1.19 (0.89 to 1.59) | 0.92 (0.71 to 1.20) |
| Children living in household | | | | |
| No children | (ref) | (ref) | (ref) | (ref) |
| Children under 6 years old | 0.82 (0.57 to 1.19) | 0.93 (0.63 to 1.36) | 1.02 (0.69 to 1.49) | 0.91 (0.65 to 1.27) |
| Children 6–11 years old | 0.76 (0.43 to 1.33) | 0.79 (0.44 to 1.45) | 0.89 (0.49 to 1.62) | 1.58 (0.91 to 2.73) |
| Children 12–19 years old | 1.04 (0.57 to 1.90) | 0.96 (0.50 to 1.84) | 0.88 (0.45 to 1.72) | 0.97 (0.54 to 1.75) |
| Trust in public information | | | | |
| Don't trust | (ref) | (ref) | (ref) | (ref) |
| Trust | 1.64 (1.19 to 2.25)** | 1.16 (0.84 to 1.61) | 1.27 (0.91 to 1.77) | 0.72 (0.53 to 0.96)* |
| Trust in unofficial information | | | | |
| Don't trust | (ref) | (ref) | (ref) | (ref) |
| Trust | 1.17 (0.88 to 1.57) | 1.75 (1.30 to 2.37)** | 1.80 (1.33 to 2.43)** | 0.91 (0.69 to 1.21) |
| Shopping in everyday life | | | | |
| No | (ref) | (ref) | (ref) | (ref) |
| Yes | 0.63 (0.42 to 0.94)* | 0.55 (0.36 to 0.83)** | 0.54 (0.36 to 0.82)** | 1.62 (1.09 to 2.43)* |
| Cooking in everyday life | | | | |
| By others | (ref) | (ref) | (ref) | (ref) |
| By themselves | 1.10 (0.81 to 1.51) | 1.33 (0.95 to 1.85) | 1.40 (1.01 to 1.96) | 0.88 (0.65 to 1.19) |

** and * indicate statistical significance at the 0.01 and 0.05 levels, respectively. (ref) means reference group for statistical comparison. GM: genetically modified.

The high school biology education targeted in this study constituted part of Japan's basic education system and could be considered relatively early education. The duration of biology education in three-year high schools in Japan is typically 12 or 24 months, which is a longer period of study than other learning options, such as reading pamphlets or attending seminars, which are the methods used in previous interventions [9–11]. We assumed that learning basic biology for a year or more in students' late teens in a high school setting would influence their aversion to GM foods; however, our results suggest that basic biology education alone, provided as children mature, is not sufficient to compensate for people's resistance to what they consider the risks of GM foods and may not be able to influence their attitudes toward GM foods. In contrast, in terms of post-high school education, less resistance was observed among those with post-graduate degrees, especially those who specialized in the sciences, which suggests that continuing education on GM foods and technology may be necessary to reduce aversion.

Previous studies have shown that age influences the acceptance of GM foods [20, 40]. Three education curricula were compared in this study. The specific education curriculum corresponded to the age of the respondents, as curricula are changed approximately every 10 years in Japan. Our results showed no significant differences in knowledge in biology knowledge test scores according to educational curriculum/age, but respondents in their 30s and 40s had a lower intention to consume more GM foods than those in their 20s. In particular, it appeared that those in their 40s had a low acceptance of GM foods, especially animal GM foods. However, this resistance may be because, as people age, there may be other, more relevant factors that affect their daily lives, such as marriage, childbirth, and health conditions. Another factor to consider is familiarity with GM food, which was very different between the 20s, 30s, and 40s groups. When GM food first appeared in Japan in 1996, those in the 40s group were already adults, while those in the 20s group had just been born. Therefore, distinguishing the effect of other factors, such as marriage and familiarity, with GM food from that of educational curricula differentiation is difficult.

One possible reason for the higher intention to eat GM foods among the 20s group and the higher resistance in the 40s group could be the time since studying biology. In this case, continuing educational methods may promote a greater understanding of GM foods. Furthermore, it is possible that information acquired during the participant's post-high school life maybe more influential.

The acceptance rate of GM vegetables, fish, and meat in our study was approximately 30%. Compared to other countries, Japan has a low acceptance of GM food [3], which is consistent with our results. According to the PCA results, Japanese people may perceive the uncertainty of GM foods as their most risky aspect; Japanese people tend to have strong uncertainty avoidance [3], which could lead them to reject GM foods.

Regarding factors other than education that could influence GM food acceptance, as household income increased, acceptance of GM foods decreased. This suggests that households with sufficient income do not need to choose GM foods and are unlikely to consume them. In addition, acceptance of GM foods was higher among those who said they trusted information from unofficial sources, such as social networking services, rather than from public institutions. This suggests that some sources took advantage of people's interest and appealed to their emotions. Regarding lifestyle factors, self-shopping was also found to influence acceptance of GM foods. These overall results suggest that the consumption of GM foods is considered in the context of daily life and cannot be separated from it. Future research should examine how important life events, such as marriage and childbirth, affect attitudes toward GM foods and what lifestyle factors influence people's acceptance of GM foods.

There are several limitations to this study. First, the study was conducted online; thus, a selection bias may be present, as the questionnaire was distributed to the company's participant pool. However, the company is one of the top three market research firms in Japan, and measures have been taken to eliminate bias as much as possible in the selection of respondents used in Japanese market research. Second, the knowledge required to understand the impact of GM foods on the human body was assessed by nine questions, but these are minimal measures and need to be examined for validity in the future. In addition, the knowledge was measured only from the perspective of the impact on the human body, and it is necessary to evaluate the participants' knowledge from broader perspectives than just the direct impact of GM foods on the human body, such as that of the environment. Finally, rather than attempting to reduce the resistance to GM foods through education, it is necessary to consider the effectiveness of providing information in daily life or along with life events, which should be considered in future research.

## Conclusion

This is the first study in Japan to examine how the high school biology curriculum affects acceptance of or resistance to GM foods. Three groups of former students from three different educational curricula and three biology education levels were compared. Results showed that high school biology education did not directly affect acceptance or resistance to GM foods, although those who received a higher level of biology education had greater knowledge of GM technologies and foods. However, post-graduate education was not associated with acceptance, but with less rejection, suggesting that continuing education may be necessary. In addition, the influence of life experience after graduation was recognized, which suggests the need to provide GM food-related information relevant to one's stage in life.

## Supporting information

**S1 Fig. Distribution of biology knowledge test score according to biology education levels.**
(TIF)

**S1 Table. Differences in biology textbook content according to text version and level.**
(DOCX)

**S2 Table. Questions on basic knowledge of DNA and digestion.**
(DOCX)

**S1 Dataset.**
(CSV)

## Acknowledgments

The authors would like to acknowledge the support from the office staff of the Department of Public Health, Health Management and Policy, and Nara Medical University, as well as Editage's editorial support.

## Author Contributions

**Conceptualization:** Akihiro Mine, Sawako Okamoto, Tomoya Myojin, Miki Hamada, Tomoaki Imamura.

**Data curation:** Akihiro Mine, Tomoya Myojin, Miki Hamada.

**Formal analysis:** Akihiro Mine, Tomoya Myojin.

**Funding acquisition:** Sawako Okamoto, Tomoaki Imamura.

**Investigation:** Akihiro Mine, Sawako Okamoto, Tomoya Myojin, Miki Hamada, Tomoaki Imamura.

**Methodology:** Akihiro Mine, Sawako Okamoto, Miki Hamada, Tomoaki Imamura.

**Project administration:** Akihiro Mine, Sawako Okamoto, Tomoya Myojin, Tomoaki Imamura.

**Resources:** Akihiro Mine, Tomoaki Imamura.

**Software:** Akihiro Mine, Tomoaki Imamura.

**Supervision:** Sawako Okamoto, Tomoya Myojin, Tomoaki Imamura.

**Validation:** Akihiro Mine, Sawako Okamoto, Tomoya Myojin.

**Visualization:** Akihiro Mine, Sawako Okamoto.

**Writing – original draft:** Akihiro Mine.

**Writing – review & editing:** Akihiro Mine, Sawako Okamoto, Tomoya Myojin, Tomoaki Imamura.

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
