## [Decision Letter · Decision Letter 0]

17 Nov 2022

PONE-D-22-25380Basic biology education in high school and acceptance of genetically modified food in JapanPLOS ONE

Dear Dr. mine,

Thank you for submitting your manuscript to PLOS ONE. After careful consideration, we feel that it has merit but does not fully meet PLOS ONE’s publication criteria as it currently stands. Therefore, we invite you to submit a revised version of the manuscript that addresses the points raised during the review process. Please submit your revised manuscript by Jan 01 2023 11:59PM. If you will need more time than this to complete your revisions, please reply to this message or contact the journal office at plosone@plos.org. Please include the following items when submitting your revised manuscript:A rebuttal letter that responds to each point raised by the academic editor and reviewer(s). You should upload this letter as a separate file labeled 'Response to Reviewers'.A marked-up copy of your manuscript that highlights changes made to the original version. You should upload this as a separate file labeled 'Revised Manuscript with Track Changes'.An unmarked version of your revised paper without tracked changes. You should upload this as a separate file labeled 'Manuscript'.

We look forward to receiving your revised manuscript.

Kind regards,

Muhammad Shahzad Aslam, Ph.D.,M.Phil., Pharm-D

Academic Editor

PLOS ONE

Journal Requirements:

2. Please provide additional details regarding participant consent. In the ethics statement in the Methods and online submission information, please ensure that you have specified what type you obtained (for instance, written or verbal, and if verbal, how it was documented and witnessed). If your study included minors, state whether you obtained consent from parents or guardians. If the need for consent was waived by the ethics committee, please include this information

Reviewers' comments:

Reviewer's Responses to Questions

**Comments to the Author**

1. Is the manuscript technically sound, and do the data support the conclusions?

Reviewer #1: Yes

Reviewer #2: Partly

Reviewer #3: Yes

Reviewer #4: Yes

Reviewer #5: Partly

2. Has the statistical analysis been performed appropriately and rigorously? 

Reviewer #1: Yes

Reviewer #2: I Don't Know

Reviewer #3: Yes

Reviewer #4: Yes

Reviewer #5: Yes

3. Have the authors made all data underlying the findings in their manuscript fully available?

Reviewer #1: Yes

Reviewer #2: No

Reviewer #3: No

Reviewer #4: Yes

Reviewer #5: Yes

4. Is the manuscript presented in an intelligible fashion and written in standard English?

Reviewer #1: Yes

Reviewer #2: Yes

Reviewer #3: Yes

Reviewer #4: Yes

Reviewer #5: Yes

5. Review Comments to the Author

Reviewer #1: General Comments

In order to investigate how basic biology education in high school affects acceptance of genetically modified foods, this questionnaire survey was conducted among the Japanese general public in their 20s to 40s. The authors used a questionnaire to assess the education program's impact on the acceptance of genetically modified foods. The authors have carefully conducted the analysis, stratifying the participants according to their educational programs and education. The topic of this paper, the acceptance of GM foods, has various stakeholders and will require careful neutral discussion. The authors seemed to attempt to discuss this topic from a neutral perspective. However, if so, a few passages could have misled the reader.

Major Comments.

1) 4. Discussion p. 22, lines 301-307.

The authors' position on GM foods is unclear, but the reviewer supposed the authors have an assumption that teaching high school biology is a way to promote GM foods. On the contrary, the purpose of teaching high school biology about GM foods may not be to encourage the acceptance or refusal of GM foods but to provide individuals with the basic knowledge to make appropriate judgments. Could you please clarify the discussion here more?

2) 4. Discussion p.22, lines 307-310.

The discussion here also reads as an assumption that high school biology education is a means to promote GM foods. It would be necessary to clarify to whom the authors are offering this discussion.

3) 4. Discussion p.22, lines 320-324.

This paragraph seems to imply that one's resistance to GM foods will decrease if one understands GM foods. However, this claim would be misleading if the authors adopt a neutral perspective toward GM foods.

4) 4. Discussion p.23, lines 340-342.

Some documentary films have raised the issue of the environmental impact of pesticides dedicated to GM crops and their industrial structure; is it not necessary to evaluate the participants' perceptions from broader perspectives than just the direct impact of GM foods on the human body?

Minor comments

5) 2.1 Data Collection

This survey is considered an exploratory study since the sample size design is not described. Such a survey is generally expected to collect as many respondents as possible relative to the population. Please add a description of the rationale for this study's sample size.

6) Table 3

Since the survey appears to be taking an exploratory approach, the reviewer would suggest that the authors also present the distribution of scores in the Supplemental figure.

Reviewer #2: The authors are trying to examine the impact of basic biology education on people’s acceptance of GM food by comparing individual experiences of biology education in a high school. I have no doubt that the purpose of this study is valuable, and the authors’ challenge should be highly appreciated. In this manuscript, however, their claims and hypotheses are not appropriately tested and thus are not explained in a convincing manner. I have listed my specific concerns below.

1. Major point

In this study, the independent variable regarding the biology education curriculum is identified by the age of participants, so it is difficult to distinguish whether its effect on the dependent variable (acceptance of GM food) is due to differences in curriculum or differences in experiences with GM foods outside of the high school education.

Specifically, GM foods first appeared in Japan in 1996, the year of the birth those who were 20 years old as of 2016, when this survey was conducted. On the other hand, those in their 40s at the time of this survey were already adults in 1996. The labeling of some foods as genetically modified was only mandated in 2001, and information and knowledge about GM foods did not become widespread until later. This means that the extent to which survey respondents were exposed to or familiar with GM foods when they were young varied greatly depending on their generation.

In addition, as the present authors state in Discussion, “as age increases after their high school education, there may be more relevant factors that affect respondents in their daily lives, such as marriage, childbirth, and health conditions (p.22)”. Therefore, it is very likely that different generations have different factors other than the content of their biology education curriculum, which may have different effects on their acceptance of GM food.

In order to determine whether or not there is an effect of biological education in high schools, researchers should carefully consider about their survey design and methodology of data analysis and try to eliminate the possibility of alternative interpretations as much as possible. For example, one may want to pay attention to the interaction between age (education curriculum) and education level in high school and compare samples with different degrees of exposure to high school biology education within the same generation. It would be also useful to examine the effect of age (education curriculum) after controlling for various surplus variables that could be related to their attitude toward GM foods. However, it appears that such efforts have not been fully made in this study. While the former may be possible to verify through additional analysis with the current data set, the latter would require a new survey with additional appropriate questions.

Also, the authors should provide an explanation of the history and background of GM foods in Japan, in Introduction of the manuscript. As mentioned above, it is closely related to the generation of the surveyed population.

2. Other points

2-1. The authors state that they conducted a content analysis of the textbooks and include the results in Table S1. Please specify the specific method used to identify and tabulate the categories, to ensure that the content analysis was not arbitrary.

2-2. The authors state that they measured acceptance of GM food, trust in information sources, and perception of risks and benefits on 4-point likert scale, but in their statistical analysis they converted these into binary values. Please specify the reason why they decided to make such a transformation. Does the shape of the distribution make it difficult to treat these scale items as continuous quantities?

Reviewer #3: This piece of work reveals a lot of commitment put into it and has a special touch of novelty and the English language fluent but there still remain some minor corrections to be made:

1. The abstract is scanty; so, go closer to the 300-word limit and really bring out your findings with p-values.

2. The cover letter should be attached.

3. The "Background and Objective" section should just read "Introduction" and the motivation to carry out the research should be well brought out here.

4. The "Method" section should read "Materials and Methods" (line 96).

5. Give a brief description of the study setting/area.

6. The authors haven't really explained to the readers how they came about a sample size of 1,594. The formula they used should be clearly explained under materials and methods.

7. Typos/grammatical errors should be corrected throughout the work. For example:

-line 199; "age" and not "edge".

-Line 109; "Firstly" and not just "First"

-Line 130; "Secondly" and not "Second"

-Line 133; "Thirdly" and not "Third"

-Line 138; "Fourthly" and not "Forth"

-Line 140; "Fifthly" and not "Fifth"

-Line 156; "Lastly" and not "Sixth"

-Line 161; "The questions on the..." and not "The questionnaire about the...".

8. Concerning the explanation made in line 135-137 and also table 1, where did you classify someone who had for example two children with ages 4 years and 7 years respectively?

9. The Results section reveals appropriate use of study objectives; however, the manuscript introduction is void of study objectives and hypothesis.

10. Table 1 makes mention only of the female sex; so, I can't tell if there were males too and if there were missing values here or not.

11. Data presentations void of graphics usage which could have added more understanding of the data presentation.

12. You don't begin the explanation with the table number i.e., referring to your explanations in line 234 and 235. The idea should come first and then the table number comes after to support your idea.

13. The data set has not been made available here for further checks.

14. No acknowledgements?

Could be fit for publication if the above inputs are done.

Reviewer #4: 13th November 2022,

The Editor,

Review of the manuscript entitled “Basic biology in high school and acceptance of genetically modified food in Japan; PONE-D-22-25380”

The article discusses a timely topic of whether the individual experience in biology education could influence people’s acceptance of genetically modified (GM) foods. The article has been well written, orderly and describes relevantly to the topic. Appropriate statistical procedures have been utilized, and have identified potential limitations of the study for further improvements, which can be highlighted as positive remarks of the present manuscript. Only minor concerns are highlighted herewith.

Minor comments

1. The article must be processed following the journal style for both format and the references. Pay attention to the English grammar errors. Some of the mistakes have been highlighted here.

2. Does the online questionnaire survey cover entire Japan or only the regional population of a particular urban region? Should it be mentioned in the discussion section?

3. In the materials and methods section, please support the statistical procedures with appropriate references.

4. Since the outcome of the study demonstrate that education alone may not be sufficient enough to consumer willingness to utilize GM foods, how the future education programs should improve to increase the consumer's attitude toward GM foods? Authors input can be important for both policy makers and future research studies.

English corrections

5. Line 27: Lay people; remove the gap between two words.

6. Line 62: “change food have increased”; replace have with has.

7. Line 63: “Curriculum”; change to curricula.

8. Line 85: Add “was” after GM food.

9. Line 91: Trust were embedded; change “were” to “was.

10. Line 116: Comprising of those; Remove “of”.

11. Line 120: People from all ages; change “from” to “of”.

12. Line 120: ten year ranges; change ranges to range.

13. Line 139: daily food cooked; change “cooked” to “cook”.

14. Line 204: four components were; change “were” to “was”.

15. Line 213: Differences of the Content; change “of” to “in”.

16. Line 219: Difference of Knowledge; change “of” to “in”.

17. Line 244: in the intentions; change “intentions” to intention.

18. Line 263: Additional Analysis on the; change “on” to “of”.

19. Line 267: Add “the” between “of” and “variance”.

20. Line 268: acceptance difference by; change “difference” to “differences”.

21. Line 277: change “everyday” to “every day”.

22. Line 286: schools influence the; change “influence” to “influences”.

23. Line 290: GM foods have reported; change “have” to “has”.

24. Line 315: had the lower; change “the” to “a”.

25. Line 316: GM foods than; add “more” after “GM”.

26. Line 324: may be is a one-word “maybe”.

With the above-mentioned corrections, I propose this study as a suitable one for publishing in PLOS ONE journal.

Thank you,

Amal Senevirathne (Ph.D.),

College of Veterinary Medicine,

Chungnam National University,

South Korea.

Reviewer #5: Basic biology education in high school and acceptance of genetically modified food in Japan

The manuscript examined the impact of basic biology education on people’s acceptance of GM food by comparing biology education and educational level using online questionnaire survey in Japan. Although they compared the results in various groups, basic questionnaire results such as acceptance rate numbers are not included in the manuscript. Also, comparative analysis using statistical methods can be presented in figures with better visualization. With the test results in tables, presentation of representative results can be improved.

Please find the detailed comments below:

Line 45: show low acceptance of and -> show low acceptance of it and

Line 114-119: Do you know from when the concept of GM food was included in the Japanese biology class? Since when did the public education introduce GM food and related biotechnology in the textbook? As you categorized people based on their ages, the differences in their GM food knowledge could be found in the curriculum in school too. If the ‘problems of biotechnology’ in Table S1 include the problems related with GM food, please indicate that for the clear explanation.

Table 1. Please add ‘Male’ in the sex column and ‘rural or suburban’ in the residence column.

Line 221: The test -> The biology knowledge test

Table 4: It is hard to tell how the biology education level and biology education were measured by the table. Please explain it in the table or in the footnote.

Table 5: (ref) should be explained in footnote. For example, Ref; reference group for statistical comparison.

Line 251-261: Please check again with the line alignment. (left -> justify)

Table 6: Acceptance rate of GM food from each group can be presented first and then the results should be compared. Please adjust the table or add one more table presenting them.

Line 308: No resistance means 0%? Where is that data? Please explain here or add proper data in the manuscript for discussion.

In discussion, the acceptance rate of GM food in other countries can be included which will be informative for readers. Also, acceptance rates in various group should be discussed.

What are the main reasons for not accepting GM foods? Although these types of questions were not included in the questionnaire, please discuss this with proper references since it is related to the conclusion of your study and future directions for the acceptance of GM foods.

6. PLOS authors have the option to publish the peer review history of their article (what does this mean?). If published, this will include your full peer review and any attached files.

Reviewer #1: No

Reviewer #2: No

Reviewer #3: No

Reviewer #4: **Yes: **Amal Senevirathne

Reviewer #5: No

---

## [Author Response · Author response to Decision Letter 0]

18 Dec 2022

November 17, 2022

Dr. Muhammad Shahzad Aslam

Academic Editor

PloS One

Dear Dr. Muhammad Shahzad Aslam:

We wish to re-submit the manuscript titled “Basic biology education in high school and acceptance of genetically modified food in Japan.” The manuscript ID is PONE-D-22-25380.

We thank you and the reviewers for the time and effort you have dedicated to providing insightful feedback. The manuscript has benefited from these suggestions. We have incorporated changes into the revised manuscript that reflect the detailed suggestions you have graciously provided. We hope that our edits and the responses we provide below satisfactorily address all the issues and concerns you and the reviewers have noted. I look forward to working with you and the reviewers to move this manuscript closer to publication in PloS One.

To facilitate your review of our revisions, the following is a point-by-point response to all editor and reviewer questions and comments.

Thank you for your consideration. I look forward to hearing from you.

Sincerely,

Akihiro Mine

Nara Medical University Department of Public Health, Health Management and Policy

840 Shijo-Cho, Kashihara, Nara, 634-8521,Japan

Tel. +81-744-22-3051 Ext. 2224

Fax. +81-744-22-0037

akihiromn777@gmail.com

Responses to Dr. Muhammad Shahzad Aslam

Thank you very much for your comments. Your comments about the submission rules are valid. We revised the manuscript according to your advice. Please see our point-by-point responses below.

Comment 1: Please ensure that your manuscript meets PLOS ONE's style requirements, including those for file naming.

Response 1: I apologize for the inadvertent errors. We have reviewed the entire paper again and corrected any issues, including the file names.

Comment 2: Please provide additional details regarding participant consent. In the ethics statement in the Methods and online submission information, please ensure that you have specified what type you obtained (for instance, written or verbal, and if verbal, how it was documented and witnessed). If your study included minors, state whether you obtained consent from parents or guardians. If the need for consent was waived by the ethics committee, please include this information.

Response 2: Thank you for pointing out this omission. We have added the details of participant consent in lines 203–208 in the revised manuscript.

Comment 3: We note that the grant information you provided in the ‘Funding Information’ and ‘Financial Disclosure’ sections do not match. When you resubmit, please ensure that you provide the correct grant numbers for the awards you received for your study in the ‘Funding Information’ section.

Response 3: We apologize for the inconsistency. The correct grant numbers is the Health and Labor Sciences Research Grant #H27-Food-General-004 in 2016. Thus, we have corrected this in lines 209–210 in the revised manuscript.

Comment 4: We note that you have indicated that data from this study are available upon request. PLOS only allows data to be available upon request if there are legal or ethical restrictions on sharing data publicly. 

Response 4: Thank you for this information. We have added Supplementary file “S1 Dataset” to the revised manuscript. All data used in this study are included in this supplementary file “S1 Dataset”.

Responses to Reviewer #1:

We thank you for the thoughtful comments. As you stated in the general comment, we tried to conduct our discussion from a neutral perspective, but there were a few points that needed to be corrected. Please see our point-by-point responses below.

Comment 1: 4. Discussion p. 22, lines 301-307.

The authors' position on GM foods is unclear, but the reviewer supposed the authors have an assumption that teaching high school biology is a way to promote GM foods. On the contrary, the purpose of teaching high school biology about GM foods may not be to encourage the acceptance or refusal of GM foods but to provide individuals with the basic knowledge to make appropriate judgments. Could you please clarify the discussion here more?

Response 1: Thank you for pointing out this ambiguity. We have revised the Introduction (lines 77–85) to clarify that we hypothesized that as students progress through the basic educational levels and obtain more basic biology knowledge, their acceptance of GM food increases. Although education itself does not necessarily move people’s intention in a specific direction, the gap between objective and subjective knowledge can create resistance to GM food [1]. Therefore, when people gain more objective knowledge through education, they may become more acceptant.

Comment 2: 4. Discussion p.22, lines 307-310.

The discussion here also reads as an assumption that high school biology education is a means to promote GM foods. It would be necessary to clarify to whom the authors are offering this discussion.

Response 2: We accept your observation, and have tried to address this point throughout our paper. As we mentioned in Response 1, we do not think that high school biology education is a means to promote acceptance of GM food, but we hypothesized that in Japan, as people gain more knowledge through education, the rate of acceptance would rise as the gap between subjective and objective knowledge shrinks.

Comment 3: 4. Discussion p.22, lines 320-324.

This paragraph seems to imply that one's resistance to GM foods will decrease if one understands GM foods. However, this claim would be misleading if the authors adopt a neutral perspective toward GM foods.

Response 3: We agree with your assessment and made a modification in lines 77–85. As mentioned in Responses 1 and 2, we do not think that education will definitively lead people to a specific decision about GM foods but that in Japan, there are specific negative beliefs commonly held about GM foods that are not supported by the science and that, if people had more objective knowledge, they would be more acceptant.

Comment 4: 4. Discussion p.23, lines 340-342.

Some documentary films have raised the issue of the environmental impact of pesticides dedicated to GM crops and their industrial structure; is it not necessary to evaluate the participants' perceptions from broader perspectives than just the direct impact of GM foods on the human body?

Response 4: We agree with your suggestion. It is true that GM foods could affect the environment and that laypeople may hold this perception; we added more information in lines 405-408.

Comment 5: 2.1 Data Collection

This survey is considered an exploratory study since the sample size design is not described. Such a survey is generally expected to collect as many respondents as possible relative to the population. Please add a description of the rationale for this study's sample size.

Response 5: Thank you for pointing this out. We have previously carried out a series of studies on the perception of GM food; in one of these studies, we set a sample size of approximately 500 [2,3]. In the current study, the initial sample size was set at 500 for each educational level, but the collection rate for the 20s group was poor, and seeking more responses by invitation e-mail would have resulted in a lower response rate and selection bias. Therefore, the target size was changed to 400. The total number of invitation e-mails was 1,594.

Comment 6: Table 3

Since the survey appears to be taking an exploratory approach, the reviewer would suggest that the authors also present the distribution of scores in the Supplemental figure.

Response 6: Thank you for your suggestion. We have added S1 Figure, which shows the distribution of biology knowledge test scores according to biology education levels.

Responses to Reviewer #2:

Thank you for providing meaningful insights. As you mentioned in the general comment, our way of presenting the hypotheses was unclear, and we revised it. Please see our point-by-point responses below.

Comment 1: In this study, the independent variable regarding the biology education curriculum is identified by the age of participants, so it is difficult to distinguish whether its effect on the dependent variable (acceptance of GM food) is due to differences in curriculum or differences in experiences with GM foods outside of the high school education.

Response 1: Thank you for your comment. As you correctly point out, after high school, there may be more relevant factors that affect respondents’ daily lives as they age, such as marriage, childbirth, and health conditions. However, it is difficult to make this distinction under our present study design. We modified the discussion to better address this point in lines 371–377.

Comment 2-1: The authors state that they conducted a content analysis of the textbooks and include the results in Table S1. Please specify the specific method used to identify and tabulate the categories, to ensure that the content analysis was not arbitrary.

Response 2-1: We apologize for not including this information; we have added the following details of the content analysis in the S1 Table:

“In the textbook content analysis, we compared the numbers of instances of specific content in the textbooks published by Tokyo Shoseki Co., Ltd. We determined how many of the following concepts were included in each textbook: 

Basic information about genetics and DNA: DNA; nucleotide; base; base complementarity; genome; base sequence; genetic information; and association between gene, DNA, and genome.

Mechanism of DNA replication: DNA and chromosomes, distribution of genomic information by somatic cell division, DNA replication, semiconservative replication, DNA polymerase, and replication error.

Mechanism of gene expression: central dogma, functions of proteins in the body such as enzymes and antibodies, RNA, transcription, reverse transcription, operons, promoters, RNA polymerase, mutations, DNA polymorphisms, functions and types of RNA, sense and antisense strands of DNA, ribozymes, exons, introns, splicing, selective splicing, deoxyribose and ribose, translation, anticodon, triplet, codon, selective gene expression, regulatory proteins and cell differentiation, regulation of gene expression by hormones, regulatory proteins, and transcription start sequences.

Gene recombination: genetic recombination, restriction enzymes, DNA ligase, vectors, transformation, cloning, PCR method, DNA sequencing, gene transfer to multicellular organisms, transgenics, GFP protein, RNA interference, genetic recombination experiments using E. coli, recombinant DNA experiments using baker’s yeast, breeding, grafting, next-generation plant-breeding techniques, self-cloning, natural occurrence, methyl group, methylation, and zinc finger nuclease.

Problems of biotechnology: use of fertilized eggs for ES cells, iPS cells, human cloning, alternative organs, human genome and privacy, overuse of pesticides, and effects of introduced genes.”

Comment 3-1: The authors state that they measured acceptance of GM food, trust in information sources, and perception of risks and benefits on 4-point likert scale, but in their statistical analysis they converted these into binary values. Please specify the reason why they decided to make such a transformation. Does the shape of the distribution make it difficult to treat these scale items as continuous quantities?

Response 3-2: Thank you for this question. We originally wanted to use dichotomous responses. However, Japanese people tend to avoid extreme values and choose midpoint values instead [4]. With this in mind, we established four responses, half of which are positive and half negative, to make respondents more comfortable. After collecting all responses, we converted the four values into two, as we originally intended.

Responses to Reviewer #3:

Thank you very much for your detailed comments. Please see our point-by-point responses below.

Comment 1: The abstract is scanty; so, go closer to the 300-word limit and really bring out your findings with p-values.

Response 1: Thank you for your suggestion. We have rewritten the abstract in lines 23–45, making it more detailed.

Comment 2: The cover letter should be attached.

Response 2: Thank you; we confirm that we did submit a cover letter.

Comment 3: The "Background and Objective" section should just read "Introduction" and the motivation to carry out the research should be well brought out here.

Response 3: Thank you for this suggestion. We have renamed and revised the Introduction; in particular, we added more information about the history of GM foods in Japan in lines 56–61. Further, we have more clearly stated the study hypothesis and objective in lines 77–85.

Comment 4: The "Method" section should read "Materials and Methods" (line 96).

Response 4: Thank you for pointing this out. We have renamed the section “Materials and Methods” in line 114.

Comment 5: Give a brief description of the study setting/area.

Response 5: Thank you. In lines 118–124, we describe how people nationwide were invited to participate. The respondents’ addresses covered all of Japan’s prefectures; we regarded metropolitan areas such as Tokyo and Osaka as urban areas and other prefectures as rural areas, as shown in Table 1. We also mentioned that education is regulated to follow the same educational guidelines nationwide in Japan.

Comment 6: The authors haven't really explained to the readers how they came about a sample size of 1,594. The formula they used should be clearly explained under materials and methods.

Response 6: You have raised an important point. However, we did not use a formula, but referenced past studies. We have previously carried out a series of studies on the perception of GM food; in one of these studies, we determined a sample size of approximately 500 [2,3]. In the current study, the initial sample size was set at 500 for each education level, but the collection rate for the 20s group was poor, and seeking more responses by invitation e-mail would have resulted in a lower response rate and selection bias. Therefore, the target size was changed to 400. The total number of invitation e-mails was 1,594.

Comment 7: Typos/grammatical errors should be corrected throughout the work.

Response 7: Thank you for bringing this to our attention. We have corrected the errors in accordance with your comment in lines 129, 138, 149, 152, 157, 159, 175, 180. Regarding line 129, we revised from edge to first as intended at the beginning. Other points, we followed your advices.

Comment 8: Concerning the explanation made in line 135–137 and also table 1, where did you classify someone who had for example two children with ages 4 years and 7 years respectively?

Response 8: We apologize for the missing information. When respondents had more than one child, we counted their youngest child. This is because we thought that parents with younger children would be more sensitive to foods their children would eat. We added this information in the revised manuscript in line 156.

Comment 9: The Results section reveals appropriate use of study objectives; however, the manuscript introduction is void of study objectives and hypothesis.

Response 9: Thank you for pointing out this omission. We have clearly stated the objectives and hypothesis of this study in the Introduction, lines 77–85.

Comment 10: Table 1 makes mention only of the female sex; so, I can't tell if there were males too and if there were missing values here or not.

Response 10: We apologize for the confusion. We have added the information on male participants to Table 1.

Comment 11: Data presentations void of graphics usage which could have added more understanding of the data presentation.

Response 11: Thank you for an important perspective. We have added S1 Figure, which shows the distribution of biology knowledge test scores according to biology education levels, to increase understanding.

Comment 12: You don't begin the explanation with the table number i.e., referring to your explanations in line 234 and 235. The idea should come first and then the table number comes after to support your idea.

Response 12: In accordance with your comment, we have modified the corresponding text in the revised manuscript in lines 281–282, as well as lines 298–300.

Comment 13: The data set has not been made available here for further checks.

Response 13: We have added Supplementary file “S1 Dataset” to the revised manuscript. All data used in this study are included in this supplementary file.

Comment 14: No acknowledgements?

Response 14: We have added acknowledgements in lines 424–427.

Responses to Reviewer #4:

Thank you for your helpful comments. Please see our point-by-point responses below.

Comment 1: The article must be processed following the journal style for both format and the references. Pay attention to the English grammar errors. Some of the mistakes have been highlighted here.

Response 1: Thank you. As per your suggestion, the manuscript has been reviewed by a professional English editing service. The formatting has also been reviewed according to journal style guidelines.

Comment 2: Does the online questionnaire survey cover entire Japan or only the regional population of a particular urban region? Should it be mentioned in the discussion section?

Response 2: Thank you for your question. In lines 118–124, we describe how people nationwide were invited to participate. Respondents’ addresses covered all of Japan’s prefectures; we regarded metropolitan areas such as Tokyo and Osaka as urban areas and other prefectures as rural areas, as shown in Table 1. 

Comment 3: In the materials and methods section, please support the statistical procedures with appropriate references.

Response 3: In accordance with your comment, we have added the supporting citations for ANOVA in lines 189–190 and for logistic analysis in lines 193–195.

Comment 4: Since the outcome of the study demonstrate that education alone may not be sufficient enough to consumer willingness to utilize GM foods, how the future education programs should improve to increase the consumer's attitude toward GM foods? Authors input can be important for both policy makers and future research studies.

Response 4: Thank you for an interesting perspective. As mentioned in lines 355–363, we think that continuing education focusing on GM foods and technology may be necessary to reduce aversion.

We corrected the writing errors according to “English corrections” in lines 27, 75, 76, 103, 134, 139, 158, 232, 240, 246, 288, 315, 319, 320, 329, 340, 344, 369, 382. Only line 109 which you pointed out, we did not correct. Because the subject of this sentence is psychological factors and we thought plurals are appropriate for predicates.

Responses to Reviewer #5:

Thank you very much for thoughtful comments. Please see our point-by-point responses below.

Comment 1: Line 45: show low acceptance of and -> show low acceptance of it and

Response 1: Thank you for pointing this out; we have revised the sentence in lines 55–56. 

Comment 2: Line 114-119: Do you know from when the concept of GM food was included in the Japanese biology class? Since when did the public education introduce GM food and related biotechnology in the textbook? As you categorized people based on their ages, the differences in their GM food knowledge could be found in the curriculum in school too. If the ‘problems of biotechnology’ in Table S1 include the problems related with GM food, please indicate that for the clear explanation.

Response 2: Thank you for an interesting perspective. We only obtained textbooks corresponding to participants’ ages because access to officially approved textbook is limited and old textbooks are difficult to obtain. To compensate, we have added more detailed information of textbook content in the note for S1 Table. Please let us know if we have adequately responded to your concern.

Comment 3: Table 1. Please add ‘Male’ in the sex column and ‘rural or suburban’ in the residence column.

Response 3: In accordance with your comment, we have added these terms to Table 1.

Comment 4: Line 221: The test -> The biology knowledge test

Response 4: Thank you for pointing out this imprecise language. It has been revised throughout the manuscript in line 247.

Comment 5: Table 4: It is hard to tell how the biology education level and biology education were measured by the table. Please explain it in the table or in the footnote.

Response 5: Thank you for this suggestion. We have added the relevant information in the table footnote in Table 4.

Comment 6: Table 5: (ref) should be explained in footnote. For example, Ref; reference group for statistical comparison.

Response 6: Thank you for pointing out this omission; we have added an explanation of “(ref)” to the footnote in Table 5.

Comment 7: Line 251-261: Please check again with the line alignment. (left -> justify)

Response 7: Thank you, we have checked all text alignment.

Comment 8: Table 6: Acceptance rate of GM food from each group can be presented first and then the results should be compared. Please adjust the table or add one more table presenting them.

Response 8: Following your advice, we have added a new table (Table 5) on acceptance rate of GM food in the revised manuscript.

Comment 9: Line 308: No resistance means 0%? Where is that data? Please explain here or add proper data in the manuscript for discussion.

Response 9: I apologize for the confusion. We meant “less resistance” and have thus revised the text accordingly in line 360.

Comment 10: In discussion, the acceptance rate of GM food in other countries can be included which will be informative for readers. Also, acceptance rates in various group should be discussed.

Response 10: Thank you for this important insight. We have added some information regarding comparison of acceptance rates from other countries in lines 383–387 in the revised manuscript.

Comment 11: What are the main reasons for not accepting GM foods? Although these types of questions were not included in the questionnaire, please discuss this with proper references since it is related to the conclusion of your study and future directions for the acceptance of GM foods.

Response 11: You have certainly raised an important point; as mentioned in lines 434–438, Japanese people’s resistance of GM food may be related to their inherent uncertainty avoidance; however, this particular mechanism was not examined in this study. As you said, we hope to examine this in future research. 

1. Hwang H, Nam S-J. The influence of consumers’ knowledge on their responses to genetically modified foods. GM Crops Food. 2021;12:146–157. doi:10.1080/21645698.2020.1840911

2. Imamura T, Ogoshi K, Hanayama H. Research Project on Promoting Security of Food Safety. H21-Food-General-007 2009. Study on Social Receptivity of Genetically Modified Food (in Japanese). 2010 May 31 [cited 2022 Jul 31]. Available from: https://mhlw-grants.niph.go.jp/project/17446

3. Imamura T, Tamura K, Matsuo M. Research Project on Promoting Security of Food Safety. H24-Food-General-005 2012. Study on Social Receptivity of Genetically Modified Food (in Japanese). 2013 Jun 24 [cited 2022 Jul 31]. Available from: https://mhlw-grants.niph.go.jp/project/21950

4. Chen C, Lee S-Y, Stevenson HW. Response Style and Cross-Cultural Comparisons of Rating Scales among East Asian and North American Students. Psychological Science. 1995;6: 170–175.

---

## [Decision Letter · Decision Letter 1]

13 Jan 2023

PONE-D-22-25380R1Basic biology education in high school and acceptance of genetically modified food in JapanPLOS ONE

Dear,

Thank you for submitting your manuscript to PLOS ONE. After careful consideration, we feel that it has merit but does not fully meet PLOS ONE’s publication criteria as it currently stands. Therefore, we invite you to submit a revised version of the manuscript that addresses the points raised during the review process.

Please revise the manuscript as per corrections given by Reviewer 3

We look forward to receiving your revised manuscript.

Kind regards,

Muhammad Shahzad Aslam, Ph.D.,M.Phil., Pharm-D

Academic Editor

PLOS ONE

Journal Requirements:

Additional Editor Comments (if provided):

Please revise the manuscript as per corrections given by Reviewer 3

Reviewers' comments:

Reviewer's Responses to Questions

**Comments to the Author**

1. If the authors have adequately addressed your comments raised in a previous round of review and you feel that this manuscript is now acceptable for publication, you may indicate that here to bypass the “Comments to the Author” section, enter your conflict of interest statement in the “Confidential to Editor” section, and submit your "Accept" recommendation.

Reviewer #1: All comments have been addressed

Reviewer #2: (No Response)

Reviewer #3: (No Response)

Reviewer #4: All comments have been addressed

Reviewer #5: All comments have been addressed

2. Is the manuscript technically sound, and do the data support the conclusions?

Reviewer #1: Yes

Reviewer #2: Partly

Reviewer #3: Yes

Reviewer #4: Yes

Reviewer #5: Yes

3. Has the statistical analysis been performed appropriately and rigorously? 

Reviewer #1: Yes

Reviewer #2: Yes

Reviewer #3: Yes

Reviewer #4: Yes

Reviewer #5: Yes

4. Have the authors made all data underlying the findings in their manuscript fully available?

Reviewer #1: Yes

Reviewer #2: Yes

Reviewer #3: Yes

Reviewer #4: Yes

Reviewer #5: Yes

5. Is the manuscript presented in an intelligible fashion and written in standard English?

Reviewer #1: Yes

Reviewer #2: Yes

Reviewer #3: Yes

Reviewer #4: Yes

Reviewer #5: Yes

6. Review Comments to the Author

Reviewer #1: (No Response)

Reviewer #2: In the previous review, I noted that the independent variable regarding the biology education curriculum is identified by the age of participants, and that it is difficult to distinguish whether its effect on the dependent variable (acceptance of GM food) is due to differences in curriculum or differences in experiences with GM foods outside of the high school education.

I think this point is very important to this manuscript. I had hoped to present further analysis and thoughtful discussion of this matter, but it was only mentioned as a limitation. I must say that I was not satisfied with this response. I believe that the research question is significant and hope that you will continue your research in order to present a more convincing argument.

Reviewer #3: Thank you very much for the answers provided to the previous comments I raised during the first review. They were quite clear and I'm sure you too were proud of these additional inputs as it went a long way to further enrich your work. Despite these, there are still some minor corrections to be made which will make the work fit for publication. They are:

1) Line 114 should read "Materials and Methods" and not "Material and Methods".

2) Lines 118-124 explains more of how data collection was done and the duration of the pilot study which does not really answer my comment number 5 of the last review. What I actually meant was your readers all over the world will be interested in knowing a little more about Japan (your study area) i.e., the total population, the population composition, climate, the soil fertility status, total surface area, socio-political atmosphere etc. So maybe you create a sub-topic under "Materials and Methods" most probably the first before "Data Collection" and caption it "Description of the Study area/setting" where you briefly explain the aforementioned.

3) About the sample size as I previously mentioned in the last review (comment 6), I'm okay with the response you gave but please do this explanation as clearly as possible under "Materials and Methods" showing the link between this work and previous works because your readers will not always have to do the kind of research you directed me to in order to understand how you came about your sample size or else it remains a myth. Moreover, you should make mention of references 2 and 3 and not just 26-31 here. Also, the adjustment from 500 to 400 questionnaires and the 1122 valid responses from 1594 should be mentioned under "Materials and Methods" and not just appearing under "Results".

If you do this, I can assure you from my side that the work will be fit for publication. Thanks

Reviewer #4: Author's have sufficiently addressed questions raised during the previous review. I went through the manuscript and can be recommended for acceptance..

Reviewer #5: The authors responded to all comments and revised the manuscript.

The manuscript examined the impact of basic biology education on people’s acceptance of GM food by comparing biology education and educational level using online questionnaire survey in Japan.

7. PLOS authors have the option to publish the peer review history of their article (what does this mean?). If published, this will include your full peer review and any attached files.

Reviewer #1: **Yes: **Taichi Hatta

Reviewer #2: No

Reviewer #3: No

Reviewer #4: No

Reviewer #5: No

---

## [Author Response · Author response to Decision Letter 1]

22 Jan 2023

January 22, 2023

Dr. Muhammad Shahzad Aslam

Academic Editor

PloS One

Dear Dr. Muhammad Shahzad Aslam:

We wish to re-submit our manuscript titled “Basic biology education in high school and acceptance of genetically modified food in Japan.” The manuscript ID is PONE-D-22-25380.

We thank you and the reviewers for the time and effort you have dedicated to providing insightful feedback. The manuscript has benefited from these suggestions. We have incorporated changes into the revised manuscript that reflect the detailed suggestions you have graciously provided. We hope that our edits and the responses we have provided below satisfactorily address all the issues and concerns that you and the reviewers have noted. I look forward to working with you and the reviewers to move this manuscript closer to publication in PloS One.

To facilitate your review of our revisions, the following is a point-by-point response to all editor and reviewer questions and comments.

Thank you for your consideration. I look forward to hearing from you.

Sincerely,

Akihiro Mine

Nara Medical University Department of Public Health, Health Management and Policy

840 Shijo-Cho, Kashihara, Nara, 634-8521, Japan

Tel. +81-744-22-3051 Ext. 2224

Fax. +81-744-22-0037

akihiromn777@gmail.com

Responses to Dr. Muhammad Shahzad Aslam

Thank you very much for your comments. Following your advice, we revised the manuscript based on the suggestions made by Reviewer 3. To follow their comment, we added reference 26 in the manuscript, and the numbers of the subsequent references have been shifted accordingly. Please see the revised manuscript.

Responses to Reviewer #1:

We thank very much for the thoughtful review. 

Responses to Reviewer #2:

Thank you again for your valuable feedback. We agree with your suggestion that the independent variable regarding the biology education curriculum is identified by the age of participants, making it difficult to distinguish whether its effect on the dependent variable (acceptance of GM food) is due to differences in curriculum or differences in experiences with GM foods outside of their high school education. However, we still believed that it would be difficult to find a valid additional analysis method in the current study design. We hope to address this in our future work.

Responses to Reviewer #3:

Thank you very much for your detailed comments. Please see our point-by-point responses below.

Comment 1: Line 114 should read "Materials and Methods" and not "Material and Methods".

Response 1: Thank you for your suggestion. We have corrected this point in line 114.

Comment 2: Lines 118-124 explains more of how data collection was done and the duration of the pilot study which does not really answer my comment number 5 of the last review. What I actually meant was your readers all over the world will be interested in knowing a little more about Japan (your study area) i.e., the total population, the population composition, climate, the soil fertility status, total surface area, socio-political atmosphere etc. So maybe you create a sub-topic under "Materials and Methods" most probably the first before "Data Collection" and caption it "Description of the Study area/setting" where you briefly explain the aforementioned.

Response 2: Thank you for your careful explanation of your previous point. Following your advice, we both added more information about the duration of the pilot study in lines 138-140, and then made a sub-topic titled, “Description of the study area,” and briefly summarized information about Japan (lines 116-135).

Comment 3: About the sample size as I previously mentioned in the last review (comment 6), I'm okay with the response you gave but please do this explanation as clearly as possible under "Materials and Methods" showing the link between this work and previous works because your readers will not always have to do the kind of research you directed me to in order to understand how you came about your sample size or else it remains a myth. Moreover, you should make mention of references 2 and 3 and not just 26-31 here. Also, the adjustment from 500 to 400 questionnaires and the 1122 valid responses from 1594 should be mentioned under "Materials and Methods" and not just appearing under "Results".

Response 3: We agree with you and have added additional information about sample size and data collection in lines 146-154.

Responses to Reviewer #4:

Thank you for your thoughtful assessment and recommendation.

Responses to Reviewer #5:

Thank you very much for your acceptance.

---

## [Editor Report · Decision Letter 2]

24 Jan 2023

Basic biology education in high school and acceptance of genetically modified food in Japan

PONE-D-22-25380R2

Dear,

We’re pleased to inform you that your manuscript has been judged scientifically suitable for publication and will be formally accepted for publication once it meets all outstanding technical requirements.

Kind regards,

Muhammad Shahzad Aslam, Ph.D.,M.Phil., Pharm-D

Academic Editor

PLOS ONE
---

## [Editor Report · Acceptance letter]

27 Jan 2023

PONE-D-22-25380R2 

Basic biology education in high school and acceptance of genetically modified food in Japan 

Dear Dr. Mine:

I'm pleased to inform you that your manuscript has been deemed suitable for publication in PLOS ONE. Congratulations! Your manuscript is now with our production department. 

Kind regards, 

on behalf of

Dr. Muhammad Shahzad Aslam 

Academic Editor

PLOS ONE